# Bias Detection via Signaling

**Yiling Chen**
Harvard University
yiling@seas.harvard.edu

**Tao Lin**
Harvard University
tlin@g.harvard.edu

**Ariel D. Procaccia**
Harvard University
arielpro@g.harvard.edu

**Aaditya Ramdas**
Carnegie Mellon University
aramdas@cmu.edu

**Itai Shapira**
Harvard University
itaishapira@g.harvard.edu

## Abstract

We introduce and study the problem of detecting whether an agent is updating their prior beliefs given new evidence in an optimal way that is Bayesian, or whether they are biased towards their own prior. In our model, biased agents form posterior beliefs that are a convex combination of their prior and the Bayesian posterior, where the more biased an agent is, the closer their posterior is to the prior. Since we often cannot observe the agent's beliefs directly, we take an approach inspired by *information design*. Specifically, we measure an agent's bias by designing a *signaling scheme* and observing the actions the agent takes in response to different signals, assuming that the agent maximizes their own expected utility. Our goal is to detect bias with a minimum number of signals. Our main results include a characterization of scenarios where a single signal suffices and a computationally efficient algorithm to compute optimal signaling schemes.

## 1 Introduction

A bag contains two coins that look and feel identical, but one is a fair coin that, on a flip, comes up heads with probability 0.5, and the other is an unfair coin with probability 0.9 of heads. You reach into the bag, grab one of the coins and flip it once; it lands on heads. Since you are (hopefully) familiar with Bayes' rule, you conclude that the probability you are holding the fair coin is $\approx 0.36$. Now suppose you are offered the following deal: if you pay \$1, you get to flip the same coin again, and if it comes up heads, you will receive \$1.4. Since you now believe that the probability of heads is 0.76, you take the deal (assuming you are risk neutral) and earn 6 cents in expectation.

If, by contrast, another risk-neutral person in the same situation decides to decline the same deal, they must believe that the probability they are holding the fair coin is greater than 0.47. That is, their belief is still very close to the prior of 0.5. We think of such a person as being *biased*, in the sense that they are unwilling to significantly update their beliefs, despite evidence to the contrary.

Of course, failing to update one's beliefs about coin flips is not the end of the world. But this example serves to illustrate a broader phenomenon that, in our view, is both important and ubiquitous. In particular, the "stickiness" of prior beliefs in the face of evidence plays a role in politics — think of the controversy over Russian collusion in the 2016 US presidential election or the existence of weapons of mass destruction in Iraq in 2003. It is also prevalent in science, as exemplified by the polarized debate over the origins of the Covid pandemic [3].

Our goal in this paper is to develop algorithms that are able to *detect* bias in the form of non-Bayesian updating of beliefs. To our knowledge, we are the first to formalize and analytically address this problem, and we aim to build an initial framework that future work would build on. In the long term, we believe such algorithms could have many applications, including understanding to what

38th Conference on Neural Information Processing Systems (NeurIPS 2024).

degree the foregoing type of bias contributes to disagreement and polarization, and discounting the opinions of biased agents to improve collective decision making.

**Our approach.** The first question we need to answer is how to *quantify* bias. In this first investigation, we adopt a linear model of bias that was proposed and used as a general belief updating model in economics [8, 10, 5, 18]. If the prior is $\mu_0$ and the correct Bayesian posterior upon receiving a *signal* (or evidence) $s$ is denoted $\mu_s$, we posit that an agent with bias $w \in [0, 1]$ adopts the belief $w\mu_0 + (1 - w)\mu_s$. At the extremes, an agent with bias $w = 0$ performs perfect Bayesian updating and an agent with bias $w = 1$ cannot be convinced to budge from the prior.

The bigger conceptual question is how we can infer an agent's bias. To address it, we take an approach that is inspired by the literature on *information design* [13]. In our context, suppose that we (the *principal*) and the agent have asymmetric information: while both share a common (say public) prior about the state of the world, the principal knows the true (realized) state of the world, but the agent does not. The *principal* publicly commits to a (randomized) *signaling scheme* that specifies the probability of sending each possible signal given each possible realized state of the world. Given their knowledge of the latter, the principal draws a signal from the specified distribution and sends it to the agent. Upon receiving such a signal, the agent updates their beliefs about the state of the world (from the common prior) and then takes an *action* that maximizes their expected utility according to a given utility function. Similarly to the example we started with, it is the action taken by the agent that can (indirectly) reveal their degree of bias.

Note that the problem of estimating the exact level of bias reduces to the problem of detecting whether the agent's bias is above or below some threshold. Indeed, to estimate the level of bias to an accuracy of $\epsilon$, $\log(1/\epsilon)$ such threshold queries suffice by using binary search. The challenge, then, is to design signaling schemes that test whether bias is above or below a given threshold in the most efficient way, that is, using a minimum number of signals in expectation.

**Our results.** We design a polynomial-time algorithm that computes optimal signaling schemes, in Section 4. We first show that *constant* algorithms, which repeatedly use the same signaling scheme, are as powerful as *adaptive* algorithms, which can vary the scheme over time based on historical data (Lemma 4.1); we can therefore restrict our attention to constant algorithms. In Lemma 4.5, we establish a version of the *revelation principle* for the bias detection problem, which asserts that optimal signaling schemes need only use signals that can be interpreted as action recommendations. Finally, building on these insights, we show that the optimal solution to our problem is obtained by solving a "small" linear program (Algorithm 1 and Theorem 4.6).

In Section 5, we present a geometric characterization of optimal signaling schemes (Theorem 5.2), which sheds additional light on the performance of the algorithm. In particular, the characterization provides sufficient and necessary conditions for the testability of bias, and also identifies cases where only a single sample is needed for this task.

**Related work.** There is a significant body of *experimental* work in the social sciences aiming to explain the failure of partisans to reach similar beliefs on factual questions where there is a large amount of publicly available evidence. The fact that biased belief updating occurs is undisputed (to our knowledge), and the focus is on understanding the factors that play a role. In particular, a prominent line of work supports the (perhaps counterintuitive) hypothesis that the more cognitively sophisticated a partisan is, the more politically biased is their belief update process [16, 17, 12, 11]. These results are challenged by more recent work by Tappin et al. [19], who found that greater analytical thinking is associated with belief updates that are less biased, using an experimental design that explicitly measures the proximity of belief updates to a correct Bayesian posterior. While these studies provide empirical underpinnings for our theoretical model, their research questions are orthogonal to ours: we aim to measure the magnitude of bias regardless of its source.

Classical work in *information design* [4, 13] studies how a principal can strategically provide information to induce an agent to take actions that are beneficial for the principal, assuming a perfectly Bayesian agent. Various relaxations of the perfectly Bayesian assumption have been investigated [1, 10, 7, 5, 9, 21, 14]. The work by de Clippel and Zhang [5] is close to us, which studies biased belief update models including the linear model. However, their goal is to maximize the principal's utility with the agent's bias fully known. In our problem the agent's bias level is unknown, and the principal's goal is to infer the agent's bias level instead of maximizing their own utility. Tang and

Ho [18] present real-world experiments showing that human belief updates are close to a linear bias model, which supports our theoretical assumption.

## 2 Model

**Biased agent.** Consider a standard Bayesian setting: the relevant *state of the world* is $\theta \in \Theta$, distributed according to some known prior distribution $\mu_0$. If an agent were perfectly Bayesian, when receiving some new information ("signal") $s$ and with the knowledge of the conditional distributions $P(s|\theta)$ for all $\theta$, they would update their belief about the state of the world according to Bayes' Rule: $\mu_s(\theta) = P(\theta|s) = \frac{\mu_0(\theta)P(s|\theta)}{P(s)}$. We refer to $\mu_s$ as the true posterior belief induced by $s$. Being biased, the agent's belief after seeing $s$, denoted $\nu_s$, is a convex combination of $\mu_s$ and $\mu_0$:

$$\nu_s = w\mu_0 + (1-w)\mu_s,$$

where $w \in [0,1]$ is the unknown *bias level*, capturing the agent's inclination to retain their prior belief in the presence of new information. This linear model was proposed and adopted in economics for non-Bayesian belief updating [8, 10], in order to capture people's conservatism in processing new information and their tendency to protect their beliefs [20].

The agent can choose an action from a finite set $A$ and has a state-dependent utility function $U : A \times \Theta \to \mathbb{R}$. They receive utility $U(a, \theta)$ when taking action $a$ in state $\theta$. The agent will act according to their (biased) belief $\nu_s$ and choose an action $a$ that maximizes their expected utility:

$$a \in \arg\max_{a \in A} \mathbb{E}_{\theta \sim \nu_s}[U(a, \theta)] = \arg\max_{a \in A} \sum_{\theta \in \Theta} \nu_s(\theta)U(a, \theta).$$

In the absence of any additional information, the agent operates based on the prior belief $\mu_0$ and will select an action deemed optimal with respect to $\mu_0$. We introduce the following mild assumption to ensure the uniqueness of this action:

**Assumption 2.1.** *There is a unique action that maximizes the expected utility based on the prior belief $\mu_0$:* $|\arg\max_{a \in A}\{\sum_{\theta \in \Theta} \mu_0(\theta)U(a, \theta)\}| = 1$.

This assumption will be made throughout the paper. We denote the unique optimal action on the prior belief as $a_0 = \arg\max_{a \in A}\{\sum_{\theta \in \Theta} \mu_0(\theta)U(a, \theta)\}$, and call it the *default action*.

**Bias detection.** The principal, who knows the prior $\mu_0$ and the agent's utility function $U$, seeks to infer the agent's bias level from their action as efficiently as possible. The principal has an informational advantage — they observe the independent realizations of the state of the world at each time step. In other words, the principal knows $\theta_t$, an independent sample drawn according to $\mu_0$ at time $t$. The principal wants to design signaling schemes to strategically reveal information about $\theta_t$ to the agent, hoping to influence the agent's biased belief in a way that the agent's chosen actions reveal information about their bias level. Specifically, with a finite signal space $S$, the principal can commit to a *signaling scheme* $\pi_t : \Theta \to \Delta(S)$ at time $t$, where $\pi_t(s|\theta)$ specifies the probability of sending signal $s$ in state $\theta$ at time $t$. After seeing a signal $s_t$, drawn according to $\pi_t(s|\theta_t)$ at time $t$, the agent takes action $a_t$ that is optimal for their biased belief $\nu_{s_t}$. The principal infers information about bias $w$ from the history of signaling schemes, realized states, realized signals, and agent actions $\mathcal{H}_t = \{(\pi_1, \theta_1, s_1, a_1), \ldots, (\pi_t, \theta_t, s_t, a_t)\}$. We denote by $\Pi$ an adaptive algorithm that the principal uses to decide on the signaling scheme at time $t + 1$ based on history $\mathcal{H}_t$.

Given a threshold $\tau \in (0, 1)$, the principal wants to design $\Pi$ to answer the following question:

*Is the agent's bias level $w$ greater than or equal to $\tau$ or less than or equal to $\tau$?*[1]

As noted earlier, by answering the above threshold question, one can also estimate the bias level $w$ within accuracy $\epsilon$ by performing $\log(1/\epsilon)$ iterations of binary search. This effectively reduces the broader task of estimating $w$ to a sequence of targeted threshold checks. By employing an adaptive signaling scheme, this approach lets us approximate $w$ to any desired precision, providing an efficient solution to the bias estimation problem.

---

[1]One may want to test $w \geq \tau$ or $w < \tau$ instead. But this requires assumptions on tie-breaking when the agent has multiple optimal actions. Indifference at $w = \tau$ allows us to avoid such assumptions.

An algorithm $\Pi$ for the above question terminates as soon as it can output a deterministic answer. The number of time steps for $\Pi$ to terminate, denoted by $T_\tau(\Pi, w)$, is a random variable. The sample complexity of $\Pi$ is defined to be the expected termination time in the worst case over $w \in [0, 1]$:

**Definition 2.1** (sample complexity). *The (worst-case) sample complexity of $\Pi$ is defined as*[2]

$$T_\tau(\Pi) = \max_{w \in [0,1]} \mathbb{E}[T_\tau(\Pi, w)].$$

Our objective is to develop an algorithm $\Pi$ that can determine whether $w \geq \tau$ or $w \leq \tau$ with minimal sample complexity. Specifically, we want to solve the following minimax problem:

$$\min_\Pi \max_{w \in [0,1]} \mathbb{E}[T_\tau(\Pi, w)].$$

We say that an algorithm $\Pi$ is *constant* if it keeps using the same signaling scheme repeatedly until termination. Constant algorithms are a special case of *non-adaptive* algorithms, which may vary the signaling schemes over time but remain independent of historical data.

**Preliminaries.** We now introduce the well-known splitting lemma from the information design literature [2, 13, 15]. It relates a signaling scheme with a set of induced true posteriors for a Bayesian agent and a distribution over the set of true posteriors.

**Lemma 2.1** (Splitting Lemma, e.g., [13]). *Let $\pi$ be a signaling scheme where each signal $s \in S$ is sent with unconditional probability $\pi(s) = \sum_{\theta \in \Theta} \mu_0(\theta)\pi(s|\theta)$ and induces true posterior $\mu_s$. Then, the prior $\mu_0$ equals the convex combination of $\{\mu_s\}_{s \in S}$ with weights $\{\pi(s)\}_{s \in S}$: $\mu_0 = \sum_{s \in S} \pi(s)\mu_s$. Conversely, if the prior can be expressed as a convex combination of distributions $\mu'_s \in \Delta(\Theta)$: $\mu_0 = \sum_{s \in S} p_s\mu'_s$, where $p_s \geq 0, \sum_{s \in S} p_s = 1$, then there exists a signaling scheme $\pi$ where each signal $s$ is sent with unconditional probability $\pi(s) = p_s$ and induces posterior $\mu'_s$.*

The splitting lemma is also referred as the Bayesian consistency condition. It allows one to think about choosing a signaling scheme as choosing a set of true posteriors, $\{\mu_s\}_{s \in S}$, and a distribution over the set, $\{\pi(s)\}_{s \in S}$, in a Bayesian consistent way.

## 3 Warm-Up: A Two-State, Two-Action Example

How can the principal design a signaling scheme to learn the agent's bias level? We use a simple two-state, two-action example to demonstrate how inducing a specific true posterior belief will allow the principal to determine whether $w \geq \tau$ or $w \leq \tau$.

The two states of the world are represented as $\{\text{Good}, \text{Bad}\}$. The agent has two possible actions: Active and Passive. Taking the Passive action always yields a utility of 0, independently of the state. For the Active action, the utility is $a$ if the state is Good and $-b$ otherwise; $a, b > 0$. We use the probability of the Good state to represent a belief, so the prior is a number $\mu_0 \in [0, 1]$, which is only a slight abuse of notation. With belief $\mu \in [0, 1]$ for the Good state (and $1 - \mu$ for the Bad state), the agent's expected utility for choosing the Active action is $a\mu - b(1 - \mu) = (a + b)\mu - b$. Thus, the Active action is better than the Passive action (so the agent will take Active) if

$$(a + b)\mu - b > 0 \quad \Longleftrightarrow \quad \mu > \tfrac{b}{a+b} =: \mu^*. \tag{1}$$

Conversely, the Passive action is better if $\mu < \mu^*$. Here, $\mu^* = \frac{b}{a+b}$ is an *indifference belief* where the agent is indifferent between the two actions. We assume that the prior $\mu_0$ satisfies $0 < \mu_0 < \mu^*$, so the agent chooses the Passive action by default.

Consider the following constant signaling scheme $\pi_\tau$ with two signals $\{G, B\}$:

- If the state is Good, send signal $G$ with probability one.

- If the state is Bad, send signal $B$ with probability $\frac{\mu^* - \mu_0}{(\mu^* - \tau\mu_0)(1 - \mu_0)}$ and signal $G$ with the complement probability.

---

[2]Taking the worst case over $w \in [0, 1]$ is not overly pessimistic. As we will show in the proofs, the worst case in fact happens at $w \in [\tau - \varepsilon, \tau + \varepsilon]$ for some $\varepsilon > 0$, which makes intuitive sense. Therefore, the sample complexity can be equivalently defined as $T_\tau(\Pi) = \max_{w \in [\tau - \varepsilon, \tau + \varepsilon]} \mathbb{E}[T_\tau(\Pi, w)]$.

We will show that, by repeatedly using $\pi_\tau$, we can test whether the agent's bias $w$ is $\le \tau$ or $\ge \tau$. By Bayes' Rule, the true posterior beliefs (for the Good state) associated with the two signals are $\mu_B = 0$ (i.e., on receiving $B$, the agent knows the state is Bad for sure) and

$$\mu_G = P(\text{Good}|G) = \frac{\mu_0 \cdot \pi_\tau(G|\text{Good})}{\mu_0 \cdot \pi_\tau(G|\text{Good}) + (1-\mu_0) \cdot \pi_\tau(G|\text{Bad})} = \frac{\mu^* - \tau\mu_0}{1-\tau}.$$

Notably, the posterior $\mu_G$ satisfies the following property: if the agent's bias level $w$ is exactly equal to $\tau$, then the agent's biased belief is equal to the indifference belief:

$$\text{when } w = \tau, \qquad \nu_G = \tau\mu_0 + (1-\tau)\mu_G = \mu^*.$$

We also note the inequality $\mu_0 < \mu^* < \mu_G$. As a result, if the agent's bias level $w$ is greater than $\tau$, then the biased belief will be smaller than $\mu^*$, and otherwise the opposite is true:

$$\text{for } w > \tau, \quad w\mu_0 + (1-w)\mu_G < \mu^*; \qquad \text{for } w < \tau, \quad w\mu_0 + (1-w)\mu_G > \mu^*.$$

By Equation (1), this means that the agent will take the Passive action if $w > \tau$, and the Active action if $w < \tau$ (on receiving $G$). Therefore, by observing which action is taken by the agent when signal $G$ is sent, we can immediately tell whether $w \le \tau$ or $w \ge \tau$. This leads to the following:

**Theorem 3.1.** *In the two-state, two-action example, for any threshold $\tau \in [0, \frac{1-\mu^*}{1-\mu_0}]$, the above constant signaling scheme $\pi_\tau$ can test whether the agent's bias $w$ satisfies $w \le \tau$ or $w \ge \tau$: specifically, whenever the signal $G$ is sent,*

- *if the agent takes action* Active*, then $w \le \tau$,*
- *if the agent takes action* Passive*, then $w \ge \tau$.*

*The sample complexity of this scheme is $\frac{\mu^* - \mu_0}{\mu_0(1-\tau)} + 1$, which increases with $\tau$.*

*Proof.* The range $\tau \in [0, \frac{1-\mu^*}{1-\mu_0}]$ ensures that the probability $\pi_\tau(B|\text{Bad}) = \frac{\mu^* - \mu_0}{(\mu^* - \tau\mu_0)(1-\mu_0)}$ is in $[0,1]$. The two items in the theorem follow from the argument before the theorem statement. The sample complexity is equal to the expected number of time steps until a $G$ signal is sent, which is a geometric random variable with success probability $P(G) = \mu_0\pi_\tau(G|\text{Good}) + (1-\mu_0)\pi_\tau(G|\text{Bad}) = \frac{\mu_0(1-\tau)}{\mu^* - \tau\mu_0}$. So the sample complexity is equal to the mean $\frac{1}{P(G)} = \frac{\mu^* - \mu_0}{\mu_0(1-\tau)} + 1$. $\qquad\square$

The *main intuition* behind this result is that in order to test whether $w \ge \tau$ or $w \le \tau$, we design a signaling scheme where certain signals induce posteriors that make the agent *indifferent between two actions if the agent's bias level is exactly $\tau$*. Then, the action actually taken by the agent will directly reveal whether $w \ge \tau$ or $w \le \tau$. Such signals are *useful* signals, but not all signals are necessarily useful. The sample complexity is then determined by the total probability of useful signals. This intuition will carry over to computing the optimal signaling scheme for the general case in Section 4.

Finally, we remark that using the constant signaling scheme $\pi_\tau$ constructed above to test $w \ge \tau$ or $w \le \tau$ is in fact the optimal adaptive algorithm, according to the results we will present in Section 4. So, the minimal sample complexity to test whether $w \ge \tau$ or $w \le \tau$ in this two-state, two-action example is exactly $\frac{\mu^* - \mu_0}{\mu_0(1-\tau)} + 1$ as shown in Theorem 3.1.

## 4 Computing the Optimal Signaling Scheme in the General Case

In this section, we generalize the initial observations from the previous section to the case with any number of actions and states and general utility function $U$. We will show how to compute the optimal algorithm (signaling scheme) to test the agent's bias level. There are three key ingredients. First, we prove that we can use a constant signaling scheme. Second, we develop a "revelation principle" to further simplify the space of signaling schemes. Building on these two steps, we show that the optimal signaling scheme can be computed by a linear program.

### 4.1 Optimality of Constant Signaling Schemes

In this subsection, we show that adaptive algorithms are no better than constant algorithms for the problem of testing whether $w \ge \tau$ or $w \le \tau$. Therefore, to find the algorithm with minimal sample complexity, we only need to consider constant algorithms/signaling schemes.

**Lemma 4.1.** *Fix $\tau \in (0, 1)$. For the problem of testing whether $w \geq \tau$ or $w \leq \tau$, the sample complexity of any adaptive algorithm is at least that of the optimal constant algorithm (i.e., using a fixed signaling scheme repeatedly).*

To prove this lemma, we introduce some notations. For any action $a \in A \setminus \{a_0\}$, define vector

$$c_a = (c_{a,\theta})_{\theta \in \Theta} = \big(U(a_0, \theta) - U(a, \theta)\big)_{\theta \in \Theta} \in \mathbb{R}^{|\Theta|}, \tag{2}$$

whose components are the utility differences between the default action $a_0$ and any other action $a$ at different states $\theta \in \Theta$. Let $R_{a_0} \subseteq \Delta(\Theta)$ be the region of beliefs under which the agent strictly prefers $a_0$ over any other action:

$$R_{a_0} = \big\{\mu \in \Delta(\Theta) \mid \forall a \in A \setminus \{a_0\}, c_a^\top \mu > 0\big\}.$$

It is the intersection of $|A| - 1$ open halfspaces with the probability simplex $\Delta(\Theta)$. As the agent strictly prefers $a_0$ at the prior $\mu_0$, we have $\mu_0 \in R_{a_0}$. The boundary of this region, $\partial R_{a_0}$, is the set of beliefs where the agent is indifferent between $a_0$ and at least one other action $a \in A \setminus \{a_0\}$ and $a_0$ and $a$ are both (weakly) better than any other action:

$$\partial R_{a_0} = \big\{\mu \in \Delta(\Theta) \mid \exists a \in A \setminus \{a_0\}, c_a^\top \mu = 0 \text{ and } \forall a' \in A \setminus \{a_0\}, c_{a'}^\top \mu \geq 0\big\}. \tag{3}$$

Lastly, the exterior of $R_{a_0}$, denoted as $\text{ext} R_{a_0}$, comprises the set of beliefs where the agent strictly prefers not to choose $a_0$:

$$\text{ext} R_{a_0} = \Delta(\Theta) \setminus (R_{a_0} \cup \partial R_{a_0}) = \big\{\mu \in \Delta(\Theta) \mid \exists a \in A \setminus \{a_0\}, c_a^\top \mu < 0\big\}.$$

Given a signaling scheme $\pi$, we classify its signals into three types based on the location of the biased belief associated with the signal with respect to the region $R_{a_0}$.

**Definition 4.1.** *Let $\tau \in (0, 1)$ be a parameter. Let $s \in S$ be a signal from a signaling scheme $\pi$, with associated true posterior $\mu_s$ and $\tau$-biased posterior $\mu_s^\tau = \tau \mu_0 + (1 - \tau)\mu_s$. We say $s$ is*

- *an **internal signal** if $\mu_s^\tau \in R_{a_0}$;*
- *a **boundary signal** if $\mu_s^\tau \in \partial R_{a_0}$;*
- *an **external signal** if $\mu_s^\tau \in \text{ext} R_{a_0}$.*

The above classification helps to formalize the idea of whether a signal is "useful" for bias detection. A boundary signal is useful because the action taken by the agent after receiving a boundary signal immediately tells whether $w \geq \tau$ or $w \leq \tau$:

**Lemma 4.2.** *When a boundary signal is realized, the agent's action immediately reveals whether $w \geq \tau$ or $w \leq \tau$. Specifically, if the agent chooses action $a_0$, then $w \geq \tau$; otherwise, $w \leq \tau$.*

*Proof.* If the agent's bias level satisfies $w < \tau$, then the biased belief $\nu_s = w \mu_0 + (1 - w)\mu_s$ must be inside $R_{a_0}$ (because $\mu_s^\tau = \tau \mu_0 + (1 - \tau)\mu_s$ is on the boundary of $R_{a_0}$ and $\mu_0 \in R_{a_0}$), so the agent strictly prefers the default action $a_0$. If $w > \tau$, then the biased belief $\nu_s$ is outside of $R_{a_0}$, so the agent will not take action $a_0$. □

An external signal might also be useful in revealing whether $w \geq \tau$ or $w \leq \tau$ if the agent is indifferent between some actions $a_1, a_2$ other than $a_0$ at the $\tau$-biased belief $\mu_s^\tau$. However, the following lemma shows that, in such cases, we can always modify the signaling scheme to turn the external signal into a boundary signal. This modification will increase the total probability of useful signals and hence reduce the sample complexity. The proof of this lemma is in Appendix A.1.

**Lemma 4.3.** *Suppose $\Pi$ is an adaptive algorithm that uses signaling schemes with internal, boundary, and external signals. Then, there exists another adaptive algorithm $\Pi'$ with equal or lower sample complexity that employs only signaling schemes with internal and boundary signals.*

An internal signal, on the other hand, is not useful for testing $w \geq \tau$ or $w \leq \tau$, for the following reason. For an internal signal, the biased belief with bias level $\tau$, $\mu_s^\tau$, lies inside $R_{a_0}$. Since $R_{a_0}$ is an open region, there must exist a small number $\varepsilon > 0$ such that when the agent has bias level $w = \tau + \varepsilon$ or $\tau - \varepsilon$, the biased belief with bias level $w$, $w \mu_0 + (1 - w)\mu_s$, is also inside the region $R_{a_0}$, so the agent will take action $a_0$. As the agent takes $a_0$ under both $w = \tau + \varepsilon$ and $\tau - \varepsilon$, we cannot distinguish these two cases, so this signal is not helpful in determining $w \geq \tau$ or $w \leq \tau$. The following lemma formalizes the idea that internal signals are not useful:

**Lemma 4.4.** *To test whether $w \geq \tau$ or $w \leq \tau$, any adaptive algorithm that uses signaling schemes with boundary and internal signals cannot terminate until a boundary signal is sent.*

*Proof of Lemma 4.1.* By Lemma 4.3, the optimal adaptive algorithm only uses signaling schemes with boundary and internal signals. By Lemma 4.4, the algorithm cannot terminate until a boundary signal is sent. By Lemma 4.2, the algorithm terminates when a boundary signal is sent. We conclude that the termination time of any adaptive algorithm cannot be better than the constant algorithm that keeps using the signaling scheme that maximizes the total probability of boundary signals. □

## 4.2 Revelation Principle

To compute the optimal constant signaling scheme, we need another technique that is similar to the *revelation principle* in the information design literature [13, 6]. The revelation principle says that, in some information design problems, it is without loss of generality to consider only "direct" signaling schemes where signals are recommendations of actions for the agent, that is, the signal space is $S = A$, and when the principal sends signal $a$, it should be optimal for the agent to take action $a$ given the posterior belief induced by signal $a$.

Unlike classical information design problems where the agent is unbiased, our problem involves a biased agent, so we need a different revelation principle: the signals are still action recommendations, but when the principal sends signal $a$, action $a$ is optimal for an agent with bias level exactly $\tau$; moreover, if $a \neq a_0$, then an agent with bias level $\tau$ will be indifferent between $a$ and $a_0$. This insight is formalized in the following lemma:

**Lemma 4.5** (revelation principle for bias detection). *Let $\pi$ be an arbitrary signaling scheme that can test $w \geq \tau$ or $w \leq \tau$. Then, there exists another signaling scheme $\pi'$ that can do so with signal space $S = A$ such that:*

*(1) Given signal $a \in A$, action $a$ is an optimal action for any agent with bias level $w = \tau$.*
*(2) Given signal $a \in A \setminus \{a_0\}$, actions $a$ and $a_0$ are both optimal for any agent with bias level $w = \tau$. As a corollary, if the agent's bias level $w < \tau$, then the agent strictly prefers $a$ over $a_0$; and if $w > \tau$, then the agent strictly prefers $a_0$ over any other actions.*
*(3) The sample complexity satisfies $T_\tau(\pi') \leq T_\tau(\pi)$.*

In the above signaling scheme $\pi'$, every $a \in A \setminus \{a_0\}$ is a boundary signal (Definition 4.1), which is useful for testing bias: given signal $a \in A \setminus \{a_0\}$, if the agent takes action $a_0$, then it must be $w \geq \tau$; otherwise $w \leq \tau$. The signal $a_0$ is internal and not useful for determining $w \geq \tau$ or $w \leq \tau$. So, the sample complexity of $\pi'$ is equal to the expected time steps until a signal in $A \setminus \{a_0\}$ is sent.

The idea behind Lemma 4.5 is *combination of signals*. Suppose there is a signaling scheme that can determine whether $w \geq \tau$ or $w \leq \tau$ with a signal space larger than $A$. There must exist two signals $s$ and $s'$ under which the agent is indifferent between $a_0$ and some action $a \neq a_0$ if the agent's bias level is exactly $\tau$. We can then combine the two signals into a single signal $s''$ under which the agent remains indifferent between $a_0$ and $a$, yielding a new signaling scheme with a smaller signal space. Repeating this can reduce the signal space to size $|A|$. See Appendix A.3 for the full proof.

## 4.3 Algorithm for Computing the Optimal Signaling Scheme

Finally, we present an algorithm to compute the optimal (minimal sample complexity) signaling scheme to test whether $w \geq \tau$ or $w \leq \tau$. The revelation principle in the previous subsection ensures that we only need a direct signaling scheme where signals are action recommendations. The optimal direct signaling scheme turns out to be solvable by a linear program, detailed in Algorithm 1. In the linear program, the constraint in Equation (5) ensures that whenever the principal recommends action $a \in A$, it is optimal for an agent with bias level $\tau$ to take action $a$; this satisfies condition (1) in the revelation principle (Lemma 4.5). The indifference constraint (Equation (5)) ensures that when the recommended action $a$ is not $a_0$, an agent with bias level $\tau$ is indifferent between $a$ and $a_0$; this satisfies condition (2) in the revelation principle. The objective (Equation (4)) is to maximize the probability of useful signals (those in $A \setminus \{a_0\}$), hence minimize the sample complexity.

**Theorem 4.6.** *Algorithm 1 finds a constant signaling scheme for testing $w \geq \tau$ or $\leq \tau$ that is optimal among all adaptive signaling schemes. The sample complexity of the optimal signaling scheme is $1/p^*$, where $p^*$ is the optimal objective value in Equation (4).*

---

**Algorithm 1:** Linear program to compute the optimal signaling scheme

---

**Input** : prior $\mu_0$, utility function $U$, and the parameter $\tau \in (0, 1)$
**Variable:** signaling scheme $\pi$, consisting of conditional probabilities $\pi(a|\theta)$ for $a \in A, \theta \in \Theta$

Denote $\Delta U(a, a', \theta) = U(a, \theta) - U(a', \theta)$. Solve the following linear program:

$$\text{Maximize} \quad \sum_{a \in A \setminus \{a_0\}} \sum_{\theta \in \Theta} \pi(a|\theta)\mu_0(\theta) \tag{4}$$

subject to:

$$\begin{cases} \text{Optimality of } a \text{ over other actions: } \forall a \in A, \forall a' \in A \setminus \{a\} \\ \qquad \sum_{\theta \in \Theta} \pi(a|\theta) \cdot \mu_0(\theta) \Big[ (1-\tau)\Delta U(a, a', \theta) + \tau \sum_{\theta' \in \Theta} \mu_0(\theta')\Delta U(a, a', \theta') \Big] \geq 0; \tag{5} \\ \text{Indifference between } a \text{ and } a_0: \forall a \in A \setminus \{a_0\}, \\ \qquad \sum_{\theta \in \Theta} \pi(a|\theta) \cdot \mu_0(\theta) \Big[ (1-\tau)\Delta U(a, a_0, \theta) + \tau \sum_{\theta' \in \Theta} \mu_0(\theta')\Delta U(a, a_0, \theta') \Big] = 0; \tag{6} \\ \text{Probability distribution constraints: } \forall \theta \in \Theta, \\ \qquad \sum_{a \in A} \pi(a|\theta) = 1 \quad \text{and} \quad \forall a \in A, \ \pi(a|\theta) \geq 0. \end{cases}$$

---

Using the above optimal signaling scheme, whenever the principal recommends an action $a$ other than $a_0$, the agent's action immediately reveals whether $w \geq \tau$ or $w \leq \tau$: if the agent indeed follows the recommendation or takes any other action than $a_0$, then the bias must be small ($w \leq \tau$); if the agent takes $a_0$ instead, the bias must be large ($w \geq \tau$). Thus, the expected sample complexity is equal to the expected number of iterations until a signal in $A \setminus \{a_0\}$ is sent, which is $1/p^*$[3].

The linear program in Algorithm 1 has a polynomial size in $|A|$ (the number of actions) and $|\Theta|$ (the number of states), so it is a polynomial-time algorithm. The solution $p^*$ depends on the geometry of the problem instance and does not seem to have a closed-form expression.

The remainder of this section proves Theorem 4.6. The proof requires an additional lemma:

**Lemma 4.7.** *Given a signaling scheme* $\pi = (\pi(a|\theta))_{a \in A, \theta \in \Theta}$ *and an agent's bias level* $w$, *after signal* $a$ *is sent, the agent strictly prefers action* $a_1$ *over* $a_2$ *under the biased belief if and only if:*

$$\sum_{\theta \in \Theta} \pi(a|\theta) \cdot \mu_0(\theta) \Big[ (1-w)\Delta U(a_1, a_2, \theta) + w \sum_{\theta' \in \Theta} \mu_0(\theta')\Delta U(a_1, a_2, \theta') \Big] > 0.$$

*Proof.* The agent's biased belief under signal $a$ and bias level $w$ is given by $(1 - w)\frac{\mu_0(\theta)\pi(a|\theta)}{\sum_{\theta' \in \Theta} \mu_0(\theta')\pi(a|\theta')} + w\mu_0(\theta), \ \forall \theta \in \Theta$. The condition for the agent to strictly prefer $a_1$ over $a_2$ is that the expected utility under the biased belief when choosing $a_1$ is greater than that of $a_2$:

$$\sum_{\theta \in \Theta} \left( (1-w)\frac{\mu_0(\theta)\pi(a|\theta)}{\sum_{\theta' \in \Theta} \mu_0(\theta')\pi(a|\theta')} + w\mu_0(\theta) \right) \Delta U(a_1, a_2, \theta) > 0,$$

where $\Delta U(a_1, a_2, \theta) = U(a_1, \theta) - U(a_2, \theta)$. Multiplying by $\sum_{\theta' \in \Theta} \mu_0(\theta')\pi(a|\theta')$, we obtain:

$$(1-w) \sum_{\theta \in \Theta} \mu_0(\theta)\pi(a|\theta)\Delta U(a_1, a_2, \theta) + w \sum_{\theta \in \Theta} \mu_0(\theta) \sum_{\theta' \in \Theta} \mu_0(\theta')\pi(a|\theta')\Delta U(a_1, a_2, \theta) > 0.$$

Factoring out the terms, this can be rewritten as:

$$\sum_{\theta \in \Theta} \pi(a|\theta)\mu_0(\theta) \left( (1-w)\Delta U(a_1, a_2, \theta) + w \sum_{\theta' \in \Theta} \mu_0(\theta')\Delta U(a_1, a_2, \theta') \right) > 0.$$

---

[3]While our focus is the expected sample complexity, we can also derive a high-probability guarantee: with $t \geq \frac{1}{p^*} \log \frac{1}{\delta}$ iterations, we can determine whether $w \geq \tau$ or $w \leq \tau$ with probability at least $1 - \delta$. This is because the probability that no useful signal is sent after $t$ iterations is at most $(1-p^*)^t \leq \delta$ when $t \geq \frac{1}{p^*} \log \frac{1}{\delta}$.

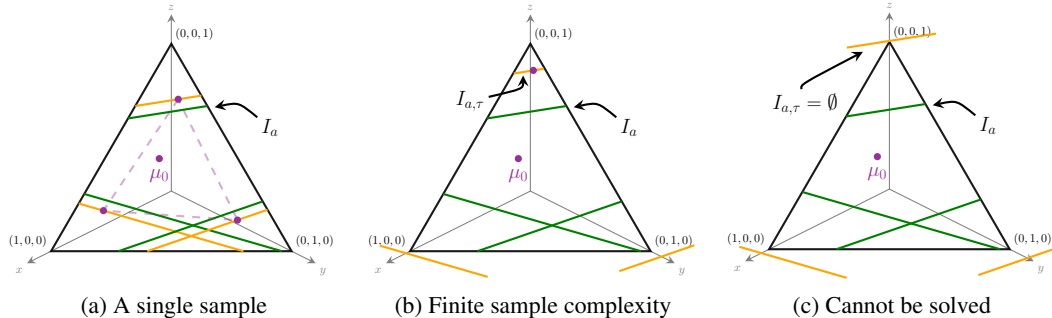

(a) A single sample       (b) Finite sample complexity       (c) Cannot be solved

Figure 1: The three qualitatively different cases for detecting the level of bias, each illustrated within a simplex over three states where $\mu_0$ is the prior belief. Each point in the simplex corresponds to an optimal action for the agent. Green curves indicate indifference between the default action $a_0$ and another action under an unbiased belief. Orange curves are translated versions of these indifference curves; a posterior on these curves means the agent's biased belief (at bias level $\tau$) aligns with the green curves. From (a) to (c), $\tau$ increases, translating the orange curves further. In Figure 1a, $\mu_0$ can be represented as a convex combination of points on the translated curves, allowing bias level detection with a single sample. In Figure 1b, only some signals are useful, requiring more than one sample in the worst case. In Figure 1c, the bias level cannot be tested against $\tau$.

This final expression is positive if and only if the agent to strictly prefer $a_1$ over $a_2$. $\qquad\square$

*Proof of Theorem 4.6.* According to Lemma 4.1 (constant algorithms are optimal) and Lemma 4.5 (revelation principle), to find an optimal adaptive algorithm we only need to find the optimal constant signaling scheme that satisfies the conditions in Lemma 4.5. We verify that the signaling scheme computed from the linear program in Algorithm 1 satisfies the conditions in Lemma 4.5:

- The optimality constraint (Equation (5)) in the linear program, together with Lemma 4.7, ensures that: whenever signal $a \in A$ is sent, action $a$ is weakly better than any other action for an agent with bias level $w = \tau$. This satisfies the first condition in Lemma 4.5.
- The indifference constraint (Equation (5)), together with Lemma 4.7, ensures that: whenever $a \in A \setminus \{a_0\}$ is sent, the agent is indifferent between action $a$ and $a_0$ if the bias level $w = \tau$. Then, by the optimality constraint (Equation (5)), we have both $a$ and $a_0$ being optimal actions. This satisfies the second condition in Lemma 4.5.

We then argue that the solution of the linear program is the optimal signaling scheme that satisfies the conditions of Lemma 4.5. According to our argument after Lemma 4.5, only the signals in $A \setminus \{a_0\}$ are useful signals, so the sample complexity is equal to the expected number of time steps until a signal in $A \setminus \{a_0\}$ is sent. The probability that a signal in $A \setminus \{a_0\}$ is sent at each time step is

$$\sum_{a \in A \setminus \{a_0\}} \pi(a) = \sum_{a \in A \setminus \{a_0\}} \sum_{\theta \in \Theta} \mu_0(\theta)\pi(a|\theta).$$

The expected number of time steps is the inverse $\frac{1}{\sum_{a \in A \setminus \{a_0\}} \sum_{\theta \in \Theta} \mu_0(\theta)\pi(a|\theta)}$ (because the number of time steps is a geometric random variable). The linear program maximizes the probability $\sum_{a \in A \setminus \{a_0\}} \sum_{\theta \in \Theta} \mu_0(\theta)\pi(a|\theta)$, so it minimizes the sample complexity. $\qquad\square$

## 5 Geometric Characterization of the Testability of Bias

To complement the algorithmic solution presented in the previous section, this section provides a geometric characterization of the bias detection problem. We identify the conditions under which testing whether $w \geq \tau$ or $w \leq \tau$ can be done in only *one* sample, in finite number of samples, or cannot be done at all (which is the scenario where the linear program in Algorithm 1 is infeasible).

By Assumption 2.1 ($a_0$ is strictly better than other actions at prior $\mu_0$), we have:

$$c_a^\top \mu_0 = \sum_{\theta \in \Theta} \mu_0(\theta)\big(U(a_0, \theta) - U(a, \theta)\big) > 0, \quad \forall a \in A \setminus \{a_0\},$$

where $c_a$ is as defined in Equation (2). Define $I_a$ as the set of indifference beliefs between action $a$ and $a_0$, which is the intersection of the hyperplane $\{x|c_a^\top x = 0\}$ and the probability simplex $\Delta(\Theta)$:

$$I_a := \{\mu \in \Delta(\Theta) \mid c_a^\top \mu = 0\}.$$

Given a parameter $\tau \in (0,1)$, for which we want to test whether $w \geq \tau$ or $w \leq \tau$, let

$$I_{a,\tau} = \{\mu \in \Delta(\Theta) \mid (1-\tau)\mu + \tau\mu_0 \in I_a\}$$

be the set of posterior beliefs for which, if the agent's bias level is exactly $\tau$, then the agent's biased belief will fall within the indifference set $I_a$.

**Lemma 5.1.** *$I_{a,\tau}$ is equal to the intersection of the probability simplex $\Delta(\Theta)$ and a translation of the hyperplane $\{x \mid c_a^\top x = 0\}$: $I_{a,\tau} = \left\{\mu \in \Delta(\Theta) \mid c_a^\top \mu = -\frac{\tau}{1-\tau} c_a^\top \mu_0\right\}$.*

The proof of this lemma is in Appendix B.1. With this representation of $I_{a,\tau}$ in hand, we can now present a geometric characterization of the testability of bias.

**Theorem 5.2** (geometric characterization). *Fix $\tau \in (0,1)$. The problem of testing $w \geq \tau$ or $w \leq \tau$*

- *Can be solved with a single sample (the sample complexity is 1) if and only if the prior $\mu_0$ is in the convex hull formed by the translated sets $I_{a,\tau}$ for all non-default actions $a \in A \setminus \{a_0\}$: i.e., $\mu_0 \in \text{ConvexHull}\left(\bigcup_{a \in A \setminus \{a_0\}} I_{a,\tau}\right)$.*
- *Can be solved (with finite sample complexity) if and only if $I_{a,\tau} \neq \emptyset$ for at least one $a \in A \setminus \{a_0\}$.*
- *Cannot be solved if $I_{a,\tau} = \emptyset$ for all $a \in A \setminus \{a_0\}$.*

Figure 1 illustrates the three cases of Theorem 5.2. In the first case, the solution of the linear program in Algorithm 1 satisfies $\sum_{a \in A \setminus \{a_0\}} \sum_{\theta \in \Theta} \pi(a|\theta)\mu_0(\theta) = 1$, meaning that useful signals are sent with probability 1, which allows us to tell whether $w \geq \tau$ or $w \leq \tau$ immediately. In the second case, the total probability of useful signals satisfies $\sum_{a \in A \setminus \{a_0\}} \sum_{\theta \in \Theta} \pi(a|\theta)\mu_0(\theta) < 1$, so the sample complexity is more than 1. In the third case, the linear program in Algorithm 1 is not feasible, so $w \geq \tau$ or $w \leq \tau$ cannot be determined; importantly, this is not a limitation of our particular algorithm, but a general impossibility in our model. The proof of Theorem 5.2 is in Appendix B.2.

## 6  Discussion

Our approach has some limitations; here we discuss the two that we view as most significant.

First, we have assumed a linear model of bias. While the linear model is common in the literature [8, 10, 5, 18], we also consider a more general model of bias (in Appendix C): as the bias level $w$ increases from 0 to 1, the agent's belief changes from the true posterior $\mu_s$ to the prior $\mu_0$ according to some general continuous function $\phi(\mu_0, \mu_s, w)$. We show that, as long as the function $\phi$ satisfies a certain single-crossing property (as $w$ increases, once the agent starts to prefer the default action $a_0$, they will not change the preferred action anymore), our results regarding the optimality of constant signaling schemes and the geometric characterization still hold, while the revelation principle and the linear program algorithm no longer work because $\phi$ is not linear. We consider it an interesting challenge to come up with more general models of bias that are still tractable, in the sense that one can efficiently design good signaling schemes with reasonable sample complexity bounds.

Second, we have assumed that the agent's prior is the same as the real prior from which states of the world are drawn. But what if the agent's prior is different? Our results directly extend to the case where the agent has a wrong, *known* prior. If the agent's prior is unknown, then our problem becomes significantly more challenging. More generally, the agent may have a private type that determines both their prior and utility and is unknown to the principal. We conjecture that testing the agent's bias in this case becomes impossible, because if different types consistently take "opposite" actions, then the actions provide no information about the agent's bias.

Despite these limitations, we view our paper as making significant progress on a novel problem that seems fundamental. Our results suggest that practical algorithms for detecting bias in belief update are within reach and, in the long term, may lead to new insights on issues of societal importance. In particular, we anticipate future research in more complex situations such as combining decisions of many experts (human or AI) after measuring and accounting for their individual biases.

## Acknowledgments

This research was partially supported by the National Science Foundation under grants IIS-2147187, IIS-2229881 and CCF-2007080; and by the Office of Naval Research under grants N00014-20-1-2488 and N00014-24-1-2704.

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

# A  Missing Proofs from Section 4

## A.1  Proof of Lemma 4.3

*Proof.* Suppose that, during its operation, $\Pi$ selects a signaling scheme $\pi$ that includes an external signal $s \in S$. By definition, for an external signal, the $\tau$-biased belief $\mu_s^\tau = \tau\mu_0 + (1-\tau)\mu_s$ is in $\text{ext} R_{a_0}$. This implies that the true posterior $\mu_s$, derived from the signaling scheme $\pi$ and the prior $\mu_0$, also lies in $\text{ext} R_{a_0}$. Consequently, the line segment connecting $\mu_s$ and $\mu_0$, represented as $\{(1-t)\mu_s + t\mu_0 \mid t \in [0,1]\}$, must intersect the boundary $\partial R_{a_0}$ at some point. Denote this intersection by $\mu^* = (1-t^*)\mu_s + t^*\mu_0 \in \partial R_{a_0}$.

We will adjust the original signaling scheme $\pi$. To do so, define $\tilde{\mu}_s$ as the belief whose $\tau$-biased version equals $\mu^*$:

$$\tau\mu_0 + (1-\tau)\tilde{\mu}_s = \mu^* \quad \Longleftrightarrow \quad \tilde{\mu}_s = \frac{(t^*-\tau)\mu_0 + (1-t^*)\mu_s}{1-\tau}.$$

Under the original signaling scheme $\pi$, according to the splitting lemma (Lemma 2.1), the prior $\mu_0$ can be represented as a convex combination of $\mu_s$ and the posteriors associated with other signals $s' \in S \setminus \{s\}$:

$$\mu_0 = p_s\mu_s + \sum_{s' \in S \setminus \{s\}} p_{s'}\mu_{s'}.$$

If we change $\mu_s$ to $\tilde{\mu}_s$, then we obtain a new convex combination (this is valid because $\tilde{\mu}_s$ is on the line segment from $\mu_s$ to $\mu_0$):

$$\mu_0 = \tilde{p}_s\tilde{\mu}_s + \sum_{s' \in S \setminus \{s\}} \tilde{p}_{s'}\mu_{s'},$$

where

$$\tilde{p}_s = \frac{p_s}{1 - t^* + t^*p_s} \quad \text{and} \quad \forall s' \in S \setminus \{s\}, \ \tilde{p}_{s'} = \frac{1 - t^*}{1 - t^* + t^*p_s}p_{s'}.$$

Then, by the splitting lemma (Lemma 2.1), there exists a signaling scheme $\pi'$ with $|S|$ signals where signal $s$ induces posterior $\tilde{\mu}_s$ and other signals $s'$ induces $\mu_{s'}$. Note that the $\tau$-biased version of $\tilde{\mu}_s$ satisfies $\tau\mu_0 + (1-\tau)\tilde{\mu}_s = \mu^* \in \partial R_{a_0}$, so $s$ is a boundary signal under signaling scheme $\pi'$.

Since $s$ is a boundary signal, we can immediately tell whether $w \geq \tau$ or $w \leq \tau$ according to Lemma 4.2 when $s$ is sent and end the algorithm. If any signal $s'$ other than $s$ is sent, the induced posterior $\mu_s$ is the same as the posterior in the original signaling scheme $\pi$, so the agent will take the same action, and we can just follow the rest of the original algorithm $\Pi$. But we note that the probability of signal $s$ being sent under the new signaling scheme $\pi'$ is larger than or equal to the probability under the original signaling scheme $\pi$:

$$\tilde{p}_s = \frac{p_s}{1 - t^* + t^*p_s} \geq p_s.$$

So, in expectation, we can end the algorithm faster by using $\tilde{\pi}$ than using $\pi$. Hence, by repeating the above procedure to replace all the signaling schemes in the original algorithm $\Pi$ that use external signals, we obtain a new algorithm $\Pi'$ that only uses boundary and internal signals with smaller or equal sample complexity. $\square$

## A.2  Proof of Lemma 4.4

*Proof.* Let $\Pi$ be any adaptive algorithm using signaling schemes with boundary and internal signals. Let $\mathcal{H}_t = \{(\pi_1, \theta_1, s_1, a_1), \ldots, (\pi_t, \theta_t, s_t, a_t)\}$ be any history that can happen during the execution of $\Pi$. If no boundary signal has been sent, then every realized signal $s_k$ is an internal signal in the respective signaling scheme $\pi_k$, with the $\tau$-biased posterior satisfying $\mu_{s_k}^\tau = \tau\mu_0 + (1-\tau)\mu_{s_k} \in R_{a_0}$. Because $R_{a_0} = \{\mu \in \Delta(\Theta) \mid \forall a \in A \setminus \{a_0\}, c_a^\top\mu > 0\}$ is an open region, there must exist some $\varepsilon_k > 0$ such that the $\ell_1$-norm ball $B_{\varepsilon_k}(\mu_{s_k}^\tau) = \{\mu \in \Delta(\Theta) : \|\mu - \mu_{s_k}^\tau\|_1 \leq \varepsilon_k\}$ is a subset of $R_{a_0}$. Let $\varepsilon = \min_{k=1}^t \varepsilon_k > 0$. Then $B_\varepsilon(\mu_{s_k}^\tau) \subseteq R_{a_0}$ for every $k = 1, \ldots, t$. Suppose the agent's bias level $w$ is in the range $[\tau - \frac{\varepsilon}{2}, \tau + \frac{\varepsilon}{2}]$. Then, for every signal $s_k$, the agent's biased belief $\nu_{s_k} = w\mu_0 + (1-w)\mu_{s_k}$ satisfies:

$$\|\nu_{s_k} - \mu_{s_k}^\tau\|_1 = \|(w-\tau)(\mu_0 - \mu_{s_k})\|_1 \leq |w - \tau| \cdot \|\mu_0 - \mu_{s_k}\|_1 \leq \varepsilon.$$

This means
$$\nu_{s_k} \in B_\varepsilon(\mu_{s_k}^\tau) \subseteq R_{a_0}.$$
So, the agent should take action $a_0$ given signal $s_k$. Note that this holds for every $k = 1, \ldots, t$ and any $w \in [\tau - \frac{\varepsilon}{2}, \tau + \frac{\varepsilon}{2}]$. So we cannot determine whether $w \geq \tau$ or $w \leq \tau$ so far. We have to run the algorithm until a boundary signal is sent. $\qquad\square$

## A.3   Proof of Lemma 4.5

*Proof.* Let $\pi$ be a signaling scheme that can test whether $w \geq \tau$ or $w \leq \tau$. According to Lemma 4.3, $\pi$ can be assumed to only use boundary and internal signals. Recall that a signal $s$ is boundary if the $\tau$-biased belief $\mu_s^\tau = \tau\mu_0 + (1 - \tau)\mu_s$ lies on the boundary set $\partial R_{a_0}$. For $a \in A \setminus \{a_0\}$, let $B_a$ be the set of beliefs under which the agent is indifferent between $a$ and $a_0$ and $a$ and $a_0$ are both better than other actions:
$$B_a = \{\mu \in \Delta(\Theta) \mid c_a^\top \mu = 0 \text{ and } \forall a' \in A, c_{a'}^\top \mu \geq 0\}.$$
The boundary set $\partial R_{a_0}$ can be written as the union of $B_a$ for $a \in A \setminus \{a_0\}$:
$$\partial R_{a_0} = \bigcup_{a \in A \setminus \{a_0\}} B_a.$$

Then, we classify the boundary signals into $|A| - 1$ sets $\{S_a\}_{a \in A \setminus \{a_0\}}$ according to which $B_a$ sets their $\tau$-biased beliefs belong to: namely, the set $S_a$ contains boundary signals $s$ under which
$$\tau\mu_0 + (1 - \tau)\mu_s \in B_a.$$
We then *combine* the signals in $S_a$. Specifically, consider the normalized weighted average of the true posterior beliefs associated with the signals in $S_a$, denoted by $\mu_a$:
$$\mu_a = \sum_{s \in S_a} \frac{\pi(s)}{\sum_{s' \in S_a} \pi(s')} \mu_s.$$
Note that the $\tau$-biased version of $\mu_a$ is also in the set $B_a$ because $B_a$ is a convex set:
$$\tau\mu_0 + (1 - \tau)\mu_a = \sum_{s \in S_a} \frac{\pi(s)}{\sum_{s' \in S_a} \pi(s')} \left(\tau\mu_0 + (1 - \tau)\mu_s\right) \in B_a.$$

This means that if a signal $a$ induces true posterior $\mu_a$, then this signal is a boundary signal.

After defining $\mu_a$ as above for every $a \in A \setminus \{a_0\}$, let's consider the set of internal signals of the signaling scheme $\pi$, which we denote by $S_I$. For each internal signal $s \in S_I$, the $\tau$-biased belief satisfies
$$\tau\mu_0 + (1 - \tau)\mu_s \in R_{a_0}.$$
Similar to above, we combine all the signals in $S_I$: define $\mu_{a_0}$ to be the normalized weighted average of the posteriors associated with all internal signals:
$$\mu_{a_0} = \sum_{s \in S_I} \frac{\pi(s)}{\sum_{s' \in S_I} \pi(s')} \mu_s.$$
Then, the $\tau$-biased version of $\mu_{a_0}$ must be in $R_{a_0}$ because $R_{a_0}$ is a convex set:
$$\tau\mu_0 + (1 - \tau)\mu_{a_0} = \sum_{s \in S_I} \frac{\pi(s)}{\sum_{s' \in S_I} \pi(s')} \left(\tau\mu_0 + (1 - \tau)\mu_s\right) \in R_{a_0}.$$

This means that, if a signal induces posterior $\mu_{a_0}$, then this signal is internal.

From the splitting lemma (Lemma 2.1), we know that the convex combination of the original posteriors $\sum_{s \in S} \pi(s)\mu_s$ is equal to the prior $\mu_0$. This means that the following convex combination of the new posteriors $\{\mu_a\}_{a \in A \setminus a_0}$ and $\mu_{a_0}$ is also equal to the prior:
$$\sum_{a \in A \setminus a_0} \sum_{s' \in S_a} \pi(s')\mu_a + \sum_{s' \in S_I} \pi(s')\mu_{a_0} = \sum_{a \in A \setminus a_0} \sum_{s \in S_a} \pi(s)\mu_s + \sum_{s \in S_I} \pi(s)\mu_s = \sum_{s \in S} \pi(s)\mu_s = \mu_0$$
where the convex combination weight of $\mu_a$ is $\sum_{s' \in S_a} \pi(s')$ for every $a \in A \setminus \{a\}$ and the convex combination weight of $\mu_{a_0}$ is $\sum_{s' \in S_I} \pi(s')$. One can easily verify that the weights sum to 1. Then, by the splitting lemma (Lemma 2.1), there exists a signaling scheme $\pi'$ with signal space of size $|A|$ (so we simply denote the signal space by $A$) where each signal $a \in A$ induces posterior $\mu_a$. We show that this new signaling scheme $\pi'$ satisfies the properties in Lemma 4.5:

- Signal $a_0$ induces posterior $\mu_{a_0}$ whose $\tau$-biased version satisfies $\tau\mu_0 + (1-\tau)\mu_{a_0} \in R_{a_0}$. So, given signal $a_0$, action $a_0$ is the optimal action for an agent with bias level $\tau$.

- For each signal $a \in A \setminus \{a_0\}$, the induced posterior $\mu_a$ satisfies $\tau\mu_0 + (1-\tau)\mu_a \in B_a \subseteq \partial R_{a_0}$. So, by the definition of $B_a$, an agent with bias level $\tau$ is indifferent between actions $a$ and $a_0$ and these two actions are better than other actions. Also, this signal is a boundary signal by Definition 4.1, which satisfies the following according to Lemma 4.2: if the agent's bias level $w < \tau$, then the agent strictly prefers $a$ over $a_0$; if $w > \tau$, then the agent strictly prefers $a_0$ over $a$.

- The sample complexity of $\pi'$ is the same as $\pi$ because: (1) the sample complexity is equal to the inverse of the total probability of boundary signals (as a corollary of Lemma 4.4), and (2) the total probability of boundary signals of the two signaling schemes are the same:

$$\sum_{a \in A \setminus \{a_0\}} \pi'(a) = \sum_{a \in A \setminus \{a_0\}} \sum_{s' \in S_a} \pi(s') = \sum_{s \in \cup_{a \in A \setminus \{a_0\}} S_a} \pi(s).$$

So, $T_\tau(\pi') = T_\tau(\pi)$.

$\square$

# B    Missing Proofs from Section 5

## B.1    Proof of Lemma 5.1

*Proof.* For $\mu \in \Delta(\Theta)$, by convexity of $\Delta(\Theta)$, we have $(1-\tau)\mu + \tau\mu_0 \in \Delta(\Theta)$. Then,

$$\mu \in I_{a,\tau} \iff (1-\tau)\mu + \tau\mu_0 \in I_a \iff c_a^\top((1-\tau)\mu + \tau\mu_0) = 0$$
$$\iff (1-\tau)c_a^\top\mu + \tau c_a^\top\mu_0 = 0$$
$$\iff c_a^\top\mu = -\frac{\tau}{1-\tau}c_a^\top\mu_0.$$

$\square$

## B.2    Proof of Theorem 5.2

We first prove the first part of Theorem 5.2, then prove the the second and third parts.

### B.2.1    Proof of Part 1 of Theorem 5.2

We want to prove that $w \geq \tau$ or $w \leq \tau$ can be tested with a single sample **if and only if** the prior $\mu_0$ is in the convex hull formed by the translated sets $I_{a,\tau}$ for all non-default actions $a \in A \setminus \{a_0\}$: $\mu_0 \in \text{ConvexHull}\left( \cup_{a \in A \setminus \{a_0\}} I_{a,\tau} \right)$.

**The "if" part.**    Suppose $\mu_0 \in \text{ConvexHull}\left( \cup_{a \in A \setminus \{a_0\}} I_{a,\tau} \right)$, namely, there exist a set of positive weights $\{p_s\}_{s \in S}$ and a set of posterior beliefs $\{\mu_s\}_{s \in S}$ such that

$$\mu_0 = \sum_{s \in S} p_s \mu_s,$$

where each $\mu_s \in I_{a,\tau}$ for some $a \in A \setminus \{a_0\}$. By definition, the $\tau$-biased belief $\tau\mu_0 + (1-\tau)\mu_s$ is in the indifference set $I_a$. Recall the definition of the boundary set $\partial R_{a_0}$ (Equation (3)), which is the set of beliefs under which the agent is indifferent between $a_0$ and some other action and these two actions are better any other actions. The $\tau$-biased belief $\tau\mu_0 + (1-\tau)\mu_s \in I_a$ may or may not belong to $\partial R_{a_0}$, depending on whether $a$ and $a_0$ are better than any other actions:

- If $\tau\mu_0 + (1-\tau)\mu_s \in \partial R_{a_0}$, then $s$ is a boundary signal (by Definition 4.1) and hence useful for testing whether $w \geq \tau$ or $w \leq \tau$ (Lemma 4.2). Denote $\mu'_s = \mu_s$ in this case.

- If $\tau\mu_0 + (1-\tau)\mu_s \notin \partial R_{a_0}$, then there must exist some action $a'$ that is strictly better than $a$ and $a_0$ for the agent at the $\tau$-biased belief, hence $\tau\mu_0 + (1-\tau)\mu_s \in \text{ext}R_{a_0}$ (so $s$ is an external signal). Then, according to the argument in Lemma 4.3, we can find another belief $\mu'_s$ on the line segment between $\mu_s$ and $\mu_0$ such that the $\tau$-biased version of $\mu'_s$ lies exactly on the boundary set $\partial R_{a_0}$:

$$\tau\mu_0 + (1-\tau)\mu'_s \in \partial R_{a_0}, \quad \mu'_s = t\mu_s + (1-t)\mu_0 \text{ for some } t \in [0,1].$$

After the above discussion, we have found a $\mu'_s$ that is either equal to $\mu_s$ or on the line segment between $\mu_s$ and $\mu_0$, for every $s \in S$. So, $\mu_0$ can be written as a convex combination of $\{\mu'_s\}_{s \in S}$:

$$\mu_0 = \sum_{s \in S} p'_s \mu'_s.$$

Moreover, the $\mu'_s$ defined above satisfies $\tau\mu_0 + (1-\tau)\mu'_s \in \partial R_{a_0}$. So, a signal inducing true posterior $\mu'_s$ will be a boundary signal and useful for testing $w \geq \tau$ or $w \leq \tau$ (Lemma 4.2). Finally, by the splitting lemma (Lemma 2.1), we know that there must exist a signaling scheme $\pi'$ with signal space $S$ where each signal $s \in S$ indeed induces posterior $\mu'_s$. Such a signaling scheme sends useful (boundary) signals with probability 1. Hence, the sample complexity of it is 1.

**The "only if" part.** Suppose whether $w \geq \tau$ or $w \leq \tau$ can be tested with a single sample. This means that the optimal signaling scheme obtained from the linear program in Algorithm 1 must satisfy $\sum_{a \in A \setminus \{a_0\}} \pi(a) = \sum_{a \in A \setminus \{a_0\}} \sum_{\theta \in \Theta} \pi(a|\theta)\mu_0(\theta) = 1$, namely, the total probability of useful signals (signals in $A \setminus \{a_0\}$) is 1. Then, by the splitting lemma, the prior $\mu_0$ can be expressed as the convex combination

$$\mu_0 = \sum_{a \in A \setminus \{a_0\}} \pi(a)\mu_a$$

where $\pi(a) = \sum_{\theta \in \Theta} \mu_0(\theta)\pi(a|\theta)$ is the unconditional probability of signal $a$ and $\mu_a$ is the true posterior induced by signal $a$. Moreover, the indifference constraint (6) in the linear program ensures that the agent is indifferent between $a$ and $a_0$ upon receiving signal $a$ if the agent has bias level $\tau$: mathematically, $\tau\mu_0 + (1-\tau)\mu_a \in I_a$. This means $\mu_a \in I_{a,\tau}$ by definition. So, we obtain

$$\mu_0 \in \text{ConvexHull}\left( \bigcup_{a \in A \setminus \{a_0\}} I_{a,\tau} \right).$$

### B.2.2 Proof of Parts 2 and 3 of Theorem 5.2

We first prove that, if whether $w \geq \tau$ or $w \leq \tau$ can be tested with finite sample complexity, then $I_{a,\tau} \neq \emptyset$ for at least one $a \in A \setminus \{a_0\}$.

According to Lemma 4.1, if we can test whether $w \geq \tau$ or $w \leq \tau$ with finite sample complexity using adaptive algorithms, then we can do this using a constant signaling scheme. Lemma 4.3 further ensures that we can do this using a constant signaling scheme $\pi$ with only boundary and internal signals. But according to Lemma 4.4, internal signals are not useful for testing $w \geq \tau$ or $w \leq \tau$. So, the signaling scheme $\pi$ must send some boundary signal $s$ with positive probability. Let $\mu_s$ be the true posterior induced by $s$. By the definition of boundary signal, $\tau\mu_0 + (1-\tau)\mu_s \in \partial R_{a_0}$, implying that the agent is indifferent between $a_0$ and some action $a \in A \setminus \{a_0\}$ if their belief is $\tau\mu_0 + (1-\tau)\mu_s$ (and $a_0$ and $a$ are better than any other actions). This means $\tau\mu_0 + (1-\tau)\mu_s \in I_a$, so $\mu_s \in I_{a,\tau}$ by definition. Hence, $I_{a,\tau} \neq \emptyset$.

We then prove the opposite direction: if $I_{a,\tau} \neq \emptyset$ for at least one $a \in A \setminus \{a_0\}$, then whether $w \geq \tau$ or $w \leq \tau$ can be tested with finite sample complexity.

Let $a_1 \in A \setminus \{a_0\}$ be an action for which $I_{a_1,\tau} \neq \emptyset$. We claim that:

**Claim B.1.** *There exists a state $\theta_1 \in \Theta$ for which the agent weakly prefers action $a_1$ over action $a_0$ if the true posterior is state $\theta_1$ with probability 1 and the agent has bias level $\tau$. In notation, let $e_{\theta_1} \in \Delta(\Theta)$ be the vector whose $\theta_1$th component is 1 and other components are 0. The agent weakly prefers action $a_1$ over action $a_0$ under belief $\tau\mu_0 + (1-\tau)e_{\theta_1}$.*

*Proof.* Suppose on the contrary that no such state $\theta_1$ exists. Then the agent strictly prefers $a_0$ over $a_1$ under belief $\tau\mu_0 + (1-\tau)e_\theta$ for every state $\theta \in \Theta$. This implies that, for any belief $\mu \in \Delta(\Theta)$,

the agent should also strictly prefer $a_0$ over $a_1$ under the belief $\tau\mu_0 + (1 - \tau)\mu$, due to linearity of the agent's utility with respect to the belief. The agent strictly preferring $a_0$ over $a_1$ implies $\tau\mu_0 + (1 - \tau)\mu \notin I_a$, so $\mu$ cannot be in $I_{a,\tau}$ by definition. This holds for any $\mu \in \Delta(\Theta)$, so $I_{a,\tau} = \emptyset$, a contradiction. $\qquad\square$

Let $\theta_1$ be the state in the above claim. The prior $\mu_0$ can be trivially written as the convex combination of $e_{\theta_1}$ and $e_\theta$ for other states $\theta$:

$$\mu_0 = \mu_0(\theta_1)e_{\theta_1} + \sum_{\theta \in \Theta \setminus \{\theta_1\}} \mu_0(\theta)e_\theta.$$

Since the agent does not prefer $a_0$ under belief $\tau\mu_0 + (1 - \tau)e_{\theta_1}$, the belief $\tau\mu_0 + (1 - \tau)e_{\theta_1}$ cannot be in the region $R_{a_0}$. The prior $\mu_0$ is in the region $R_{a_0}$. Consider the line segment connecting $e_{\theta_1}$ and the prior $\mu_0$. There must exist a point $\mu' = te_{\theta_1} + (1 - t)\mu_0$ on the line segment such that the $\tau$-biased belief $\tau\mu_0 + (1 - \tau)\mu'$ lies exactly on the boundary of $R_{a_0}$. Clearly, the prior can also be written as a convex combination of $\mu'$ and $e_\theta$ for $\theta \in \Theta \setminus \{\theta_1\}$:

$$\mu_0 = p'\mu' + \sum_{\theta \in \Theta \setminus \{\theta_1\}} p'_\theta e_\theta.$$

Then by the splitting lemma (Lemma 2.1), there exists a signaling scheme with $|\Theta|$ signals where one signal induces posterior $\mu'$ and the other signals induce posteriors $\{e_\theta\}_{\theta \in \Theta \setminus \{\theta_1\}}$. In particular, the signal inducing $\mu'$ is a boundary signal since $\tau\mu_0 + (1 - \tau)\mu' \in \partial R_{a_0}$ by construction. By Lemma 4.2, that signal is useful for testing $w \geq \tau$ or $w \leq \tau$. When that signal is sent (which happens with positive probability $p' > 0$ at each time step), we can tell $w \geq \tau$ or $w \leq \tau$. This finishes the proof.

The two directions proved above together prove the parts 2 and 3 of Theorem 5.2.

## C    A More General Bias Model

We define a more general model of biased belief than the linear model. The agent's bias is captured by some function $\phi : \Delta(\Theta) \times \Delta(\Theta) \times [0, 1] \to \Delta(\Theta)$. Given prior $\mu_0 \in \Delta(\Theta)$, true posterior $\mu_s \in \Delta(\Theta)$, and bias level $w \in [0, 1]$, the agent has biased belief $\phi(\mu_0, \mu_s, w)$. The linear model is the special case where $\phi(\mu_0, \mu_s, w) = w\mu_0 + (1-w)\mu_s$. We make the following natural assumptions on $\phi$:

**Assumption C.1.**

- $\phi(\mu_0, \mu_s, 0) = \mu_s$ *(no bias)*, $\phi(\mu_0, \mu_s, 1) = \mu_0$ *(full bias)*.

- $\phi(\mu_0, \mu_s, w)$ *is continuous in* $\mu_0, \mu_s, w$.

We then make some joint assumptions on the bias model $\phi$ and the agent's utility function $U$. Recall that the notation $R_{a_0} = \{\mu \in \Delta(\Theta) \mid \forall a \in A \setminus \{a_0\},\ c_a^\top \mu > 0\}$ is the region of beliefs under which the agent strictly prefers action $a_0$, $\partial R_{a_0}$ is the boundary of $R_{a_0}$, and $\text{ext} R_{a_0}$ is the exterior of $R_{a_0}$ where the agent strictly not prefers $a_0$.

**Assumption C.2.** *When receiving no information, the agent will take the default action:* $\forall \mu_0 \in \Delta(\Theta), \forall w \in [0, 1]$, $\phi(\mu_0, \mu_0, w) \in R_{a_0}$.

**Definition C.1.** *We say that a posterior belief* $\mu \in \Delta(\Theta)$ *satisfies* **single-crossing** *if the curve* $\{\phi(\mu_0, \mu, w) : w \in [0, 1]\}$ *passes the boundary* $\partial R_{a_0}$ *only once: namely, there exists* $\overline{w} \in [0, 1]$ *such that*

$$\begin{cases} \forall w \in [0, \overline{w}), & \phi(\mu_0, \mu, w) \in \text{ext} R_{a_0}; \\ & \phi(\mu_0, \mu, \overline{w}) \in \partial R_{a_0}; \\ \forall w \in (\overline{w}, 1], & \phi(\mu_0, \mu, w) \in R_{a_0}. \end{cases}$$

We assume that all posteriors outside $R_{a_0}$ satisfy single-crossing, and all posteriors inside $R_{a_0}$ do not cross the boundary when the bias level varies in $[0, 1]$:

**Assumption C.3.**

- *Any $\mu \notin R_{a_0}$ satisfies single-crossing.*

- *For any $\mu \in R_{a_0}$, any $w \in [0, 1]$, $\phi(\mu_0, \mu, w) \in R_{a_0}$.*

Under the above general bias model with the stated assumptions, our results regarding the optimality of constant signaling schemes (Lemma 4.1) still holds. The geometric characterization of testability of bias (Theorem 5.2) holds after redefining some notations. Let $I_a = \{\mu \in \Delta(\Theta) \mid c_a^\top \mu = 0\}$ still be the set of beliefs where the agent is indifferent between actions $a$ and $a_0$. Let $I_{a,\tau}$ still be the set of posterior beliefs for which an agent with bias level $\tau$ will be indifferent between $a$ and $a_0$, but with a more general expression than the linear model:

$$I_{a,\tau} := \{\mu \in \Delta(\Theta) \mid \phi(\mu_0, \mu, \tau) \in I_a\}.$$

With the above definition of $I_{a,\tau}$, Theorem 5.2 still holds. We omit the proofs because they are almost identical to the proofs for the linear model.

The revelation principle (Lemma 4.5) and the linear program algorithm for computing the optimal signaling scheme (Algorithm 1 and Theorem 4.6) do not apply to the general bias model because $\phi$ is not linear. Designing an efficient algorithm to compute a good signaling scheme to test bias in a more general model is an interesting future direction.

