# OpenReview forum: "Bias Detection via Signaling"
_NeurIPS.cc/2024/Conference — NeurIPS 2024 poster_

### Official Review · Reviewer_NFob · 2024-07-12

**Soundness:** 3
**Presentation:** 3
**Contribution:** 3
**Rating:** 6
**Confidence:** 3

**Summary:**

The paper studies bias detection in the Bayesian persuasion model. In particular the paper wants to test for a threshold that represents the resistance of the receiver to change its mind and update their prior. The test is performed trough signalling schemes and observe the action performed by the receiver. The paper is concerned with finding the signalling scheme that minimizes the worst case expectation of the needed number of queries. The problem is solved by geometrical techniques.

**Strengths:**

The paper is well written and fairly easy to follow. Moreover it provides interesting answers to interesting questions. The answers goes trough geometrical characterisation and the paper helps build the intuition required for understanding the main results.

**Weaknesses:**

The main weakness in my opinion is the definition of the baseline and its poor understanding in the larger context of PAC learning.
Why the baseline is defined as such rather then considering the minimum of expected samples needed to understand if \omega\ge\tau-\epsilon or \omega \le \tau+\epsilon with probability at least 1-\delta? This is a key technical point and, while negative results such as lemma 4.4 are expected with \epsilon=0, they are not expected to hold with \epsilon>0. For example for (\epsilon,\delta)-PAC learning for best arm identification in multi armed bandit has infinite samples complexity with \epsilon=0 even in trivial cases such as all the arms have the same mean.

**Questions:**

see weaknesses

**Limitations:**

yes

---

> ### Author Rebuttal · Authors · 2024-08-06
>
> > The main weakness in my opinion is the definition of the baseline and its poor understanding in the larger context of PAC learning. Why the baseline is defined as such rather than considering the minimum of expected samples needed to understand if $\omega \ge \tau - \epsilon$ or $\omega \le \tau + \epsilon$ with probability at least $1 - \delta$? This is a key technical point and, while negative results such as lemma 4.4 are expected with $\epsilon = 0$, they are not expected to hold with $\epsilon > 0$. For example, $(\epsilon, \delta)$-PAC learning for best arm identification in multi-armed bandit has infinite sample complexity with $\epsilon = 0$ even in trivial cases such as all the arms have the same mean.
>
> This is an interesting point worth clarifying, but we disagree that it is a weakness and believe we can provide an effective rebuttal.
>
> We first note that it isn't completely clear whether you're interested in the problem of testing whether $\omega\in [\tau-\epsilon,\tau+\epsilon]$ (in which case "if" and "or" should be replaced with "and") or the "gap" problem of testing if $\omega \le \tau - \epsilon$ or $\omega \ge \tau + \epsilon$ (in which case the inequalities are flipped). We assume you have the former problem in mind, as it is more consistent with the PAC setting.
>
> Next, the problem of testing whether $\omega\in [\tau-\epsilon,\tau+\epsilon]$ involves testing whether $\omega\leq \tau+\epsilon$ *and* $\omega\geq \tau-\epsilon$, that is, it is equivalent to two instances of the exact threshold problem. This may seem unintuitive, and you may wonder whether testing other thresholds in this interval might lead to better sample complexity; that is not the case, though, because the choice of bias level is adversarial (in the same way that testing a threshold $\tau$ cannot be sped up by testing another threshold $\tau'\neq \tau$).
>
> Finally, we argue that there is no reason to include a confidence parameter. Unlike standard PAC problems (where there is some underlying distribution), the only source of randomness in our problem is the signaling scheme. Recall, also, that we show that the optimal scheme simply maximizes the probability of useful signals. If that probability is $p$, the optimal scheme needs $1/p$ samples in expectation. You could ask instead about the number of samples $t$ needed for a success probability of $1-\delta$, but given this discussion, the answer is clear: $(1-p)^t\leq \delta$, which is satisfied when $t \ge \frac{1}{p} \log \frac{1}{\delta}$.
>
> Your PAC intuition may hold true if, analogously to a stochastic multi-armed bandit problem, the bias $\omega$ is drawn from a distribution, but that's a fundamentally different model (in the same way that stochastic bandits are fundamentally different from adversarial bandits).

---

> > ### Comment · Reviewer_NFob · 2024-08-12
> >
> > I thank the authors for their reply. I understand now better the relationship with PAC learning. I believe that this discussion should be expanded and included into the paper, as also other reviewers seemed to have similar questions. I confirm my evaluation

---

> > > ### Author Response · Authors · 2024-08-12
> > >
> > > Thank you for your response. We will gladly expand this discussion and include it in the paper.

---

### Official Review · Reviewer_rhkn · 2024-07-12

**Soundness:** 3
**Presentation:** 3
**Contribution:** 3
**Rating:** 5
**Confidence:** 3

**Summary:**

This paper studies the problem of determining the bias level of an agent in updating their beliefs using signaling schemes. Specifically, they detect to what degree, the agent is updating their beliefs biased towards their own prior, or 'correctly' according to the Bayesian rule. They propose a signaling scheme to detect whether the bias level is below or above a given threshold $\tau$. The core of their algorithm design is the revelation principle, which converts to a linear program subjecting to optimality, indifference, and probability distribution constraints.

**Strengths:**

The paper is in general well-written and smooth to follow. The problem formulation and scheme design intuitions are nicely explained. The problem studied is novel. The proofs seem to be theoretically sound.

**Weaknesses:**

Although the studied problem is novel, it lacks motivation and application in the real world. The authors motivate it by connecting bias to disagreement and polarization, but they don't provide any detailed discussion. Also, currently the authors assume the principal knows the agent's prior. Then it makes no sense to "discount the opinions of biased agents to improve collective decision making" because the principal knows enough information and can solve the optimal decision solely.

**Questions:**

The problem studied in this paper is determining whether the bias is below or above a given threshold $\tau$. A closely related, and probably more common problem is directly estimating the bias level using signaling. Can the authors share their thoughts on this? For example, what's the main difficulty of this new problem? Do the authors believe (variants of) constant schemes are still optimal?

**Limitations:**

I don't see any limitations or potential negative societal impact of this work.

---

> ### Author Rebuttal · Authors · 2024-08-06
>
> > Although the studied problem is novel, it lacks motivation and application in the real world. The authors motivate it by connecting bias to disagreement and polarization, but they don't provide any detailed discussion. Also, currently the authors assume the principal knows the agent's prior. Then it makes no sense to "discount the opinions of biased agents to improve collective decision making" because the principal knows enough information and can solve the optimal decision solely.
>
> Your point is well taken that expanding our motivation would be beneficial, and we plan to do so in the revision.
>
> We disagree with the specific criticism in your comment, however; in particular, we aren't sure what you have in mind when you refer to the "optimal decision," as this is not part of our model. One way to motivate our setup is to view it as the initial interaction between a decision maker and an expert before any future solicitation of information from the expert for decisions of interest. The goal of this interaction — which can be structured as an artificial game, for example — is to estimate the expert's bias so that the expert's future opinions can be appropriately adjusted, improved or even discarded. This is conceptually analogous to the design of interactions to measure individual risk attitudes before giving subjects decision-making tasks, an approach common to economists and psychologists. Note that here the decision maker only has an informational advantage in the initial interaction.
>
> To better connect our problem to polarization, we note that one can make a distinction between two sources of polarization [1]: two people can be exposed to different sources of information (e.g., news stories that reflect opposite positions), or they can be exposed to similar sources of information but reach different conclusions. Techniques for measuring bias in our sense may help disambiguate these two sources. For example, for two people who have very different world views, if we find that they have low levels of bias according to our definition, it would increase our confidence that they were exposed to different sources of information.
>
> [1] Nika Haghtalab, Matthew O. Jackson, Ariel D. Procaccia: Belief polarization in a complex world: A learning theory perspective. Proc. Natl. Acad. Sci. USA 118(19): e2010144118 (2021).
>
> > The problem studied in this paper is determining whether the bias is below or above a given threshold $\tau$. A closely related, and probably more common problem is directly estimating the bias level using signaling. Can the authors share their thoughts on this? For example, what's the main difficulty of this new problem? Do the authors believe (variants of) constant schemes are still optimal?
>
> Our approach can be directly employed to estimate the bias up to an $\epsilon$ error, by solving the threshold problem $\log(1/\epsilon)$ times, through binary search. This requires adaptive signaling schemes and constant schemes are not sufficient. We mentioned this in line 55, but we will certainly elaborate on this point (which was brought up in some form by all reviewers) in the revision.

---

> > ### Author Response · Authors · 2024-08-12
> >
> > We would greatly appreciate your response to the rebuttal, which, in our view, effectively addresses your main concerns. If there are lingering questions, we would gladly engage in a discussion.

---

### Official Review · Reviewer_1o3u · 2024-07-13

**Soundness:** 4
**Presentation:** 4
**Contribution:** 3
**Rating:** 7
**Confidence:** 5

**Summary:**

The paper studies the problem of detecting whether an agent is updating their prior beliefs given new evidence in a Bayesian way, or  whether they are biased towards their own prior. The paper considers a setting where biased agents form posterior beliefs that are a convex combination of their prior and the Bayesian posterior (parameterized by an unknown parameter $\omega$). Given a fixed $\tau$, the paper takes an information design to detect where $\omega\ge \tau$ or $\omega\le \tau$. In particular, one can design a signaling scheme and observe that actions taken by the agents to infer the bias level $\omega$. The paper aims to minimize the number of rounds (each round deploying a signaling scheme) needed to detect whether $\omega\ge \tau$ or $\omega\le \tau$.

The main results of the paper show that (1) a fixed signaling scheme suffices to achieve minimum number of rounds used detect whether $\omega\ge \tau$ or $\omega\le \tau$; (2) a computationally efficient algorithm to compute such optimal signaling scheme for detecting whether $\omega\ge \tau$ or $\omega\le \tau$.

**Strengths:**

In practice, it is widely observed that human belief updating with uncertainty is usually not following the Bayesian manner due to various biases and oftentimes such biases remain unknown to the system. Thus, it is natural to study how to detect such human bias and thus I think the motivation of this work is strong. The paper uses a novel approach via information design to detect human bias level. This, in addition, also opens up another application of information design/Bayesian persuasion. The paper also presents that using such approach is computationally efficient (for constant number of states and actions).
The paper is also well-written and results are stated in a clear way. Overall, I think this paper is a good addition to NeurIPS.

**Weaknesses:**

There may be an immediate future direction is like given any $\varepsilon\ge 0$, what is the sample complexity of narrowing down the unknown bias level $\omega$ to be in an $\varepsilon$-interval? I feel $O(\log \frac{1}{\varepsilon})$ number of rounds using binary search should suffice (though you may need to carefully handle the randomness of realized signals).

There is a previous work "HOCMP 2021 — On the Bayesian Rational Assumption in Information Design", where that work presents some real-world behavior experiments showing that humans are updating beliefs really not in a Bayesian way, and show that a convex combination of the prior and the Bayesian posterior can better explain human belief updating. The authors may want to include HOCMP 2021 and discuss its connections.

**Questions:**

See above

---

> ### Author Rebuttal · Authors · 2024-08-06
>
> > There may be an immediate future direction is like given any $\epsilon$, what is the sample complexity of narrowing down the unknown bias level $\omega$ to be in an $\epsilon$-interval? I feel $\log \frac{1}{\epsilon}$ number of rounds using binary search should suffice (though you may need to carefully handle the randomness of realized signals).
>
> Absolutely, we allude to this in line 55, but we will certainly elaborate on this point (which was brought up in some form by all reviewers) in the revision.
>
> > There is a previous work "HOCMP 2021 — On the Bayesian Rational Assumption in Information Design", where that work presents some real-world behavior experiments showing that humans are updating beliefs really not in a Bayesian way, and show that a convex combination of the prior and the Bayesian posterior can better explain human belief updating. The authors may want to include HOCMP 2021 and discuss its connections.
>
> Thanks for the excellent pointer; this will strengthen our case for focusing on the linear model of bias (convex combination of prior and Bayesian posterior), which we currently support via references [8,10,5] in line 39.

---

> > ### Comment · Reviewer_1o3u · 2024-08-12
> >
> > Thank the authors for your response. I do not have further concerns.

---

### Official Review · Reviewer_7Pvv · 2024-07-13

**Soundness:** 3
**Presentation:** 2
**Contribution:** 2
**Rating:** 5
**Confidence:** 5

**Summary:**

The paper studies a Bayesian persuasion problem involving a biased receiver. In this model, the bias is defined by how the receiver deviates from the Bayesian posterior, which is a convex combination of the prior and the induced posterior. The authors propose algorithms to test whether this bias exceeds a fixed constant and discuss scenarios where it is possible or not to fully detect the bias level. In cases where it is possible to detect this bias level, they propose algorithms to solve the problem polynomially using direct signaling schemes.

**Strengths:**

- The problem studied in this paper is of interest to the Bayesian Persuasion community.
- The geometric characterization of the problem is interesting and clear, specifically the characterization of whether or not it is possible to learn the bias level presented in section 5.

**Weaknesses:**

- Under the assumption that the sender knows everything about the environment (prior, utilities, etc.) and only needs to determine if the bias level is larger than a constant, I think that the results are not difficult to derive. The paper presents some characterizations, such as the use of constant algorithms and the study of instances in which it is possible to determine with one sample whether the bias level is greater than a constant, or whether it is not possible even with infinitely many samples. However, these characterizations alone are not sufficient to justify accepting the paper.

- The sample complexity problem studied in this paper is quite different from the classical notion, where you are given two parameters, $\epsilon$ and $\delta$, and you want to output an $\epsilon$-optimal solution with probability $1 - \delta$. I would have found it much more interesting (and standard) to compute a confidence bound for the bias level with high probability, rather than determining if the bias is larger than a constant.

- Finally, the main theorems are not clearly stated.  For example, in the statement of Theorem 3.1, you refer to "the above signaling scheme," which requires the reader to look for this scheme on the preceding page. Additionally, in the main theorem, the final sample complexity of the optimal constant signaling scheme is not specified. This makes it challenging to have a high-level understanding of the contribution of your main theorems.

**Questions:**

- In sample complexity problems, you are usually given two parameters $\epsilon$ and $\delta$, and you want to output an $\epsilon$-optimal solution with probability $1-\delta$. In your model, this can be formulated as learning the bias up to an $\epsilon$ error with probability $1-\delta$. My question is, why not study this version of the problem? Can your approach be employed to tackle this version of the problem as well?

- Can your algorithm be extended to scenarios with a multi-type receiver?

- In the case of finite sample complexity, is it possible to achieve a better dependence with respect to $\tau$, or are your results tight?

---

> ### Author Rebuttal · Authors · 2024-08-06
>
> > The sample complexity problem studied in this paper is quite different from the classical notion, where you are given two parameters, $\epsilon$ and $\delta$, and you want to output an $\epsilon$-optimal solution with probability $1 - \delta$. I would have found it much more interesting (and standard) to compute a confidence bound for the bias level with high probability, rather than determining if the bias is larger than a constant. [...] In sample complexity problems, you are usually given two parameters $\epsilon$ and $\delta$,, and you want to output an $\epsilon$-optimal solution with probability $1-\delta$. In your model, this can be formulated as learning the bias up to an $\epsilon$ error with probability $1-\delta$. My question is, why not study this version of the problem? Can your approach be employed to tackle this version of the problem as well?
>
> We believe that we can effectively address this concern, and would be happy to add the discussion below to the paper.
>
> We first note that our approach can be directly employed to estimate the bias up to an $\epsilon$ error, by solving the threshold problem $\log(1/\epsilon)$ times, through binary search, as briefly mentioned in line 55. We will expand on this in the revised paper.
>
> Let us now explain why we did not include a confidence parameter, and why our results directly imply confidence bounds if one did want to include one. First, we note that unlike the standard PAC setting (where there is some underlying distribution), the only source of randomness in our problem is the signaling scheme. As we show, the optimal scheme maximizes the probability $p$ of obtaining useful signals. If that probability is $p$, the optimal scheme needs $1/p$ samples in expectation. You could ask instead about the number of samples $t$ needed for a success probability of $1-\delta$, but given this discussion, the answer is straightforward: $(1-p)^t\leq \delta$, which is satisfied when $t \ge \frac{1}{p} \log \frac{1}{\delta}$.
>
> In other words, our current formulation implicitly provides confidence bounds, but we agree it would be valuable to make this more explicit.
>
>
> > The main theorems are not clearly stated. For example, in the statement of Theorem 3.1, you refer to "the above signaling scheme," which requires the reader to look for this scheme on the preceding page. Additionally, in the main theorem, the final sample complexity of the optimal constant signaling scheme is not specified. This makes it challenging to have a high-level understanding of the contribution of your main theorems.
>
> For Theorem 3.1, we note that it's common to refer in a theorem statement to an algorithm that is defined previously, say "Algorithm 1." In our case, we felt it would be awkward to put the signaling scheme on lines 158-161 in an environment ("Signaling Scheme 3.1") because it's so simple. However, we are open to doing so if it would improve readability.
>
> The point about the Theorem 4.6 is more important as it may stem from a misunderstanding.  The final sample complexity of the optimal signaling is equal to $1/p^*$, where $p^*$ is the objective value of the linear program (Equation 6, Algorithm 1) and represents the probability of useful signals. This sample complexity depends on the geometry of the instance and lacks a simple closed-form solution. The main contribution of Theorem 4.6 — which builds on Lemmas 4.1-4.5 — is algorithmic. Section 4 as a whole demonstrates that the optimal signaling scheme can be computed efficiently. While Section 5 provides insights into the sample complexity (and Theorem 3.1 gives tight bounds for a special case), we emphasize that the general case where the sample complexity is finite but greater than 1 does not appear to have a more detailed, concise characterization.
>
>
> > Can your algorithm be extended to scenarios with a multi-type receiver?
>
> The concept of a "multi-type receiver" could be interpreted in various ways.
>
> If the sender knows the receiver's type in each round, we believe our results can be directly extended to this multi-type setting.
>
> However, if the sender does not know the receiver's type, we conjecture that testing the receiver's bias becomes impossible. The intuition is that if different types consistently take "opposite" actions, the receiver's observed action provides no information about their bias.
>
> > In the case of finite sample complexity, is it possible to achieve a better dependence with respect to $\tau$, or are your results tight?
>
> Our results are tight for both the two-state-two-action case (Theorem 3.1) and the general case (Theorem 4.6), as they provide the optimal signaling schemes.

---

> > ### Author Response · Authors · 2024-08-12
> >
> > We would greatly appreciate your response to the rebuttal, which, in our view, effectively addresses your main concerns. If there are lingering questions, we would gladly engage in a discussion.

---

> > > ### Comment · Reviewer_7Pvv · 2024-08-12
> > >
> > > I thank the authors for their responses. Regarding the PAC learning formulation of the problem, I suggest to the authors to better explain this aspect in the final version of the paper and also discuss the possibility of extending their model to a multi-typed receiver. I believe this would strengthen the current submission. Furthermore, since some of my concerns have been addressed in this rebuttal, I am increasing my score by one point to a borderline accept.

---

> > > > ### Author Response · Authors · 2024-08-12
> > > >
> > > > Thank you very much. We are happy to follow these suggestions.

---

### Decision · Program_Chairs · 2024-09-25

**Decision:**

Accept (poster)

**Comment:**

The paper studies the problem of detecting a bias in the way an agent updates their prior belief given new evidence. Specifically, it does so by using techniques borrowed from the information design (aka Bayesian persuasion) literature, namely, through the provision of (partially) informative signals to the agent.

All the Reviewers agree on the fact that the paper is well-written and studies an interesting problem. The results presented in the paper also allow to reach interesting conclusions. The Reviewers only raised some (minor) concerns that have been adequately addressed by the Authors in the rebuttal.

Thus, I recommend the paper to be accepted at NeurIPS, **subject to the Authors addressing all the issues raised by the Reviewers in the final version of the paper**.